# Effects of temperature on the life history traits of intermediate host snails of fascioliasis: A systematic review

**Agrippa Dube** [1]*, **Chester Kalinda**[1,2], **Tawanda Manyangadze**[1,3], **Tafadzwa Mindu**[1], **Moses John Chimbari**[1,4]

**1** School of Nursing and Public Health, College of Health Sciences, Howard College Campus, University of KwaZulu-Natal, Durban, South Africa, **2** University of Global Health Equity (UGHE), Bill and Joyce Cummings Institute of Global Health, Kigali Heights, Kigali, Rwanda, **3** Geosciences Department, School Geosciences, Disaster and Sustainable Development, Faculty of Science and Engineering, Bindura University of Science and Technology, Bindura, Zimbabwe, **4** Office of the Pro-Vice Chancellor: Academic Affairs, Research and Innovation, Great Zimbabwe University, Masvingo, Zimbabwe

* adube168@gmail.com

**Data Availability Statement:** All relevant data are within the paper.

## Abstract

### Background

The impact of climate change has led to variations in various biological processes, leading to altered transmission dynamics of infectious diseases, including snail-borne diseases (SBDs). Fascioliasis is one of the neglected zoonotic tropical snail-borne diseases caused by the trematode of the genus *Fasciola*. This review focused on laboratory experimental and model studies that evaluate the potential effect of temperature change on the ecology and biology of the intermediate host snails (IHS) of *Fasciola*.

### Methods

A literature search was conducted on Google Scholar, EBSCOhost, and PubMed databases using predefined medical subject heading terms, Boolean operators, and truncation symbols in combination with direct keywords: Fasciolosis AND Temperature, *Lymnaea* OR *Austropeplea OR Radix* OR *Galba* OR *Fossaria* OR *Pseudosuccinea* AND growth, fecundity, AND survival at the global scale. Other search terms used were (Fasciliasis AND Temperature), (*Lymnaea* AND Temperature), (*Austropeplea* AND Temperature), (*Fossaria* AND Temperature), (Galba AND Temperature), (*Pseudosuccinea* AND Temperature), and (*Radix* AND Temperature).

### Results

The final synthesis included thirty-five published articles. The studies reviewed indicated that temperature rise may alter the distribution, and optimal conditions for breeding, growth, and survival of IHS, ultimately resulting in changing the transmission dynamics of fascioliasis. The literature also confirmed that the life history traits of IHS and their interaction with the liver fluke parasites are driven by temperature, and hence climate change may have

**Funding:** This project has received funding from the European Union's Horizon 2020 Research and Innovation Program under grant agreement No 101000365. This work was supported the European Union's Horizon 2020 Research and Innovation Program to M J C. Agrippa Dube received monthly stipend from the European Union's Horizon 2020 Research and Innovation Program. The funders had no role in study design, data collection and analysis, decision to publish, or preparation of the manuscript.

**Competing interests:** The authors declare that they have no conflicts of interest.

profound outcomes on the population size of snails, parasite density, and disease epidemiology.

## Conclusion

We concluded that understanding the impact of temperature on the growth, fecundity, and survival of IHS may broaden our knowledge of the possible effects of climate change and hence inform fascioliasis control programs.

### Author summary

Fascioliasis is a neglected tropical disease caused by flatworms or liver flukes of the genus *Fasciola*. The review paper focuses on the effect of temperature on the life history traits of intermediate hosts of fascioliasis. The authors aim to explore the impact of temperature on the growth, survival, and reproduction of the host snails and the development of parasites inside intermediate hosts. Fascioliasis is considered a major public health problem in many countries, affecting both humans and livestock. The disease is transmitted through the consumption of metacercaria contaminated watercress for humans and vegetation for grazing livestock. The review paper highlights that temperature plays a crucial role in the life history traits of these intermediate hosts. The authors gathered and analyzed various studies conducted on different intermediate host snails of *Fasciola* species to investigate the relationship between temperature and the development, survival, reproduction, and infectivity of the parasites within them. Their findings reveal that the temperature can significantly influence the life history traits of these intermediate hosts. Higher temperatures generally promote faster development, increased reproduction, and higher infectivity rates. However, there is a limit to the beneficial effects of temperature, beyond which the survival and fitness of the intermediate hosts start to decline. The authors also emphasize the relevance of understanding the impact of temperature on fascioliasis in the context of climate change. As global temperatures continue to rise, the distribution and prevalence of fascioliasis may change, affecting the transmission dynamics of the disease. This knowledge is crucial for the design and implementation of effective control strategies to mitigate the impact of fascioliasis on human and animal health.

## Background

Fascioliasis is a parasitic, zoonotic, neglected tropical snail-borne disease of veterinary and public health importance [1–4]. The disease, also known as liver rot or distomatosis, is caused by the digenean trematodes *Fasciola hepatica* and *Fasciola gigantica*, with the former contributing significantly to the global burden of the disease [5–10]. *Fasciola hepatica* has the most widespread distribution and host range (from temperate to tropical regions on all continents except Antarctica). *Fasciola gigantica* is more restricted to Asia and Africa's tropical and subtropical climates [11]. Both *F. hepatica* and *F. gigantica* have a five-stage life cycle that includes an egg, miracidium, intramolluscan stages of trematodes (sporocysts, rediae, and cercariae), metacercariae, and adults [12]. The liver fluke's life cycle is complex, with definitive mammalian hosts, notably cattle, donkeys, sheep, and pigs; many free-living phases in the environment; and an intermediate molluscan host, a lymnaeid snail [13,14]. Globally, about 20 species of lymnaeid snails were identified as potential intermediate hosts of *Fasciola* spp. [15]. *Galba*

*truncatula* (also called *Lymnaea truncatula*) is involved in the transmission of *F. hepatica* in the temperate regions, while *Radix natalensis* (also called *Lymnaea natalensis*) is involved in the transmission of *F. gigantica* in the tropics [12,16]. *Pseudosuccinea columella* (also called *Lymnaea columella*), an invasive species, has become an important intermediate host of both *Fasciola* species in many regions of the world [17]. *Galba truncatula* is characterized by its amphibious lifestyle, adaption to cooler settings, and ability to tolerate drought conditions [16]. *Galba truncatula* hibernates and aestivates during the winter and drought seasons, respectively [16,18]. Understanding fascioliasis requires knowledge of its chains of transmission and IHS species. Infection in livestock occurs when animals feed on metacercariae-contaminated pastures, whereas humans can get it accidentally by eating watercress and drinking water contaminated with metacercariae, respectively [1,8,14,19–21]. In order to interrupt disease transmission, IHS should be the ultimate targets of control programs because they represent the weakest link in the disease transmission cycle [22]. The transmission of fascioliasis is affected by several abiotic and biotic factors, ranging from climatic factors such as temperature and rainfall to snail food, mammalian hosts, the latitude or altitude of the snail habitat, and the presence of vector snails and parasites [23,24]. The parasite's free-living stages thrive in warm, moist conditions, which also promote the reproduction and survival of IHS such as *G. truncatula* [8,25,26]. Because temperature and moisture influence multiple stages of the lifecycle, changing climatic conditions may influence the timing, intensity, and distribution of fascioliasis outbreaks [26].

Higher temperatures change the dynamics of trematode (fluke) disease throughout the critical stages of its life cycle by shortening the pre-patent time in the first intermediate host [14,27,28]. Climate change with milder temperatures and heavier rainfall has been predicted to increase the risk of fascioliasis [29] and this may impact on the use of triclabendazole, with its repeated use observed to led to mutants or strains of *Fasciola* that are triclabendazole resistant [29–31]. Thus, enhanced knowledge of how climate change influences host-parasite interactions is needed. This review explored how temperature affects the growth, fecundity, and survival of *Lymnaea*, *Galba*, *Pseudosuccinea*, *Austropeplea*, *Fossaria* and *Radix* snail species and the development of intramolluscan stages of the trematodes in IHS.

## Materials and methods

### Search strategy

A systematic search of the literature on Google Scholar, PubMed, and EBSCOhost databases was conducted using the following terms and Boolean operators (OR, AND): Fasciolosis AND Temperature, *Lymnaea* OR *Radix* OR *Galba* OR *Pseudosuccinea* OR *Austropeplea* OR *Fossaria* AND growth, fecundity, AND survival at the global scale. Other search terms used were (Fascioliasis AND Temperature), (*Lymnaea* AND Temperature), (*Galba* AND Temperature), (*Pseudosuccinea* AND Temperature), (*Fossaria* AND Temperature), (*Austropeplea* AND Temperature) and (*Radix* AND Temperature), Mathematical Models AND Temperature AND Fascioliasis AND Intermediate Hosts. The identified articles were initially screened by reading the titles and abstracts. Furthermore, the selected articles' references and bibliographic lists were searched for potential leads to additional relevant studies for inclusion. Full-text articles were retrieved and managed in the Endnote 20 reference manager (Clarivate Analytics, Philadelphia, PA, USA).

### Inclusion criteria

Studies were considered for inclusion in the review if they were published in peer-reviewed scientific journals and specifically reported on (1) the effect of temperature on life history traits

of IHS of *Fasciola hepatica* and *Fasciola gigantica* that cause animal and human fascioliasis; (2) the genus *Lymnaea*, *Austropeplea*, *Fossaria*, *Galba*, *Radix*, *Pseudosuccinea*; and (3) only articles written in English were considered in this study.

## Exclusion criteria

Exclusion criteria were: i) studies concerning a different parasite than *F. gigantica* or *F. hepatica*; ii) studies that do not include *Lymnaea* sp., *Fossaria sp*., *Galba sp*., *Radix sp*., *Austropeplea sp or Pseudosuccinea sp*.; iii) articles not written in English; iv) studies reporting results outside of the scope of the review question; and v) studies not reporting on the effect of temperature on life history traits of IHS.

The systematic literature search yielded a total of one thousand two hundred twenty-seven plus seventeen reports from other sources, which included abstracts, reports, books, and theses

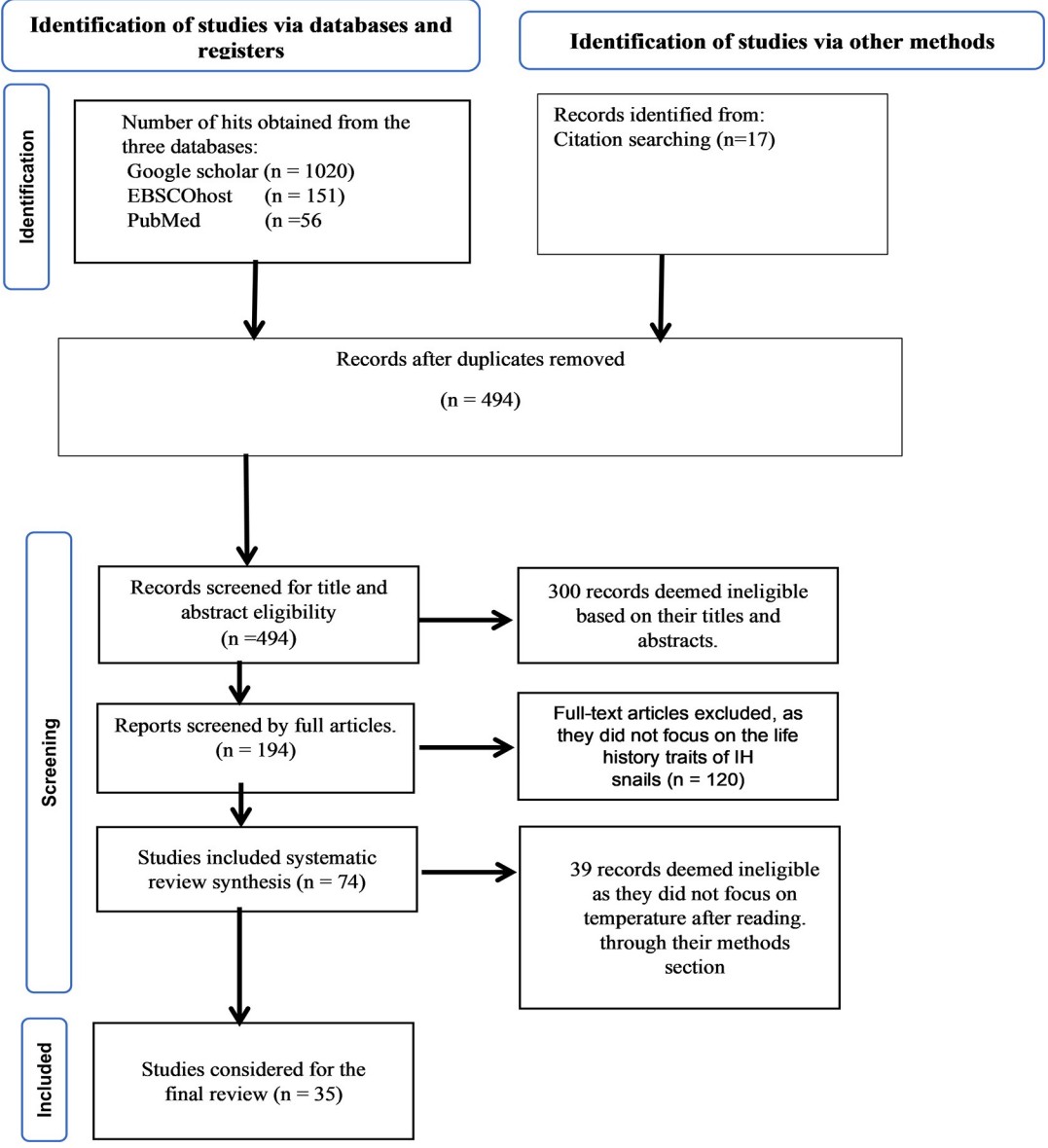

**Fig 1. PRISMA diagram shows the study selection process.**

(Fig 1). Seven hundred fifty articles were eliminated because of duplication, and three hundred articles were removed as they were deemed ineligible based on their titles and abstracts. A further one hundred and twenty articles were excluded because they did not focus on temperature despite looking at the life history traits of IHS, and thirty-nine were further dropped as they did not specifically report on temperature about the IHS *Lymnaea* spp, *Fossaria* spp, *Galba* spp, *Austropeplea* spp *Pseudosuccinea* spp, and *Radix* spp of *Fasciola* spp. Table 1 lists the thirty-five publications that were considered for this evaluation. The findings from the publications that met the criteria for the review were categorized into the following themes: temperature and fecundity, temperature and growth, temperature and survival, and temperature and parasite development. These are discussed below about potential changes in the disease risk with a possible rise in temperature.

The study selection process is shown in Fig 1.

## Results

### Temperature effect on IHS fecundity

Reviewed articles in Table 1, showed that temperature influence the time of sexual maturity and fertility in Lymnaeid snails [50]. In temperate *Galba truncatula*, reproduction rate and fecundity ceased at temperature below 5˚C. In one of the experiments involving *Galba truncatula*, it was observed that reproduction rate and fecundity ceased at a temperature below 5˚C [33,50]. In another study, it was observed that fecundity occurred under experimental conditions at temperatures as low as 10–11˚C for *Galba truncatula* [60]. The reproduction temperature range for *Galba truncatula* was observed to be 10–25˚C. The reproduction rate was high at 24˚C for *Pseudosuccinea columella*, and this was also observed to tolerate cooler, warmer, and desert temperatures [66]. However, it was observed that the reproduction rate of *Radix acuminata* in the tropics occurred optimally at a temperature range of 16–34˚C [39,40]. Furthermore, Aziz [42] and Misra [41] found that the reproduction rate of *Radix luteola* (Lamarck) and *Radix acuminata*, respectively, was higher at lower temperature ranges of 15–20˚C. In subtropical regions, 5–10˚C was the minimal temperature needed for *Pseudosuccinea columella* and *Austopeplea* (*Lymnaea*) *tomentosa* egg development and hatching [66]. It was observed that *Lymnaea stagnalis* eggs require constant temperatures between 9.9˚C and 28.0˚C for development and hatching [57]. Most studies found that snail reproduction increased within a certain temperature range, but fecundity was significantly reduced below and beyond that temperature range (Table 1).

### Temperature effect on IHS growth

Temperature has been shown to affect IHS growth [41,42,61,62]. In temperate regions, the optimal growth rate of *Lymnaea viatrix* snails was at 25˚C, and it declined at temperatures above 25˚C [52,61]. In another study, it was noted that snail growth increased between temperatures of 15˚C and 24˚C for *Lymnaea viridis* [56]. Also, the growth rate for *Lymnaea stagnalis* increased with temperature, with 25˚C being optimal [50]. At 20˚C, most *Radix peregra* snails showed significantly higher growth rates [49]. In the tropics, it was observed that the rate of growth increased with temperature from 15–30˚C for *Radix acuminata* [42]. In another experiment, it was noted that at 15–22.5˚C, juvenile *Galba truncatula* snails became active, and their growth increased rapidly [33]. In the subtropics, temperature is particularly influential in the development of *G. truncatula* and *F. hepatica*, which require temperatures ≥10˚C for growth, reproduction, and survival [62]. The studies reviewed revealed that IHS species from different regions have different thermal tolerances for growth (Table 2).

**Table 1.** Summary of laboratory experimental studies and models that assessed the effect of temperature on *Lymnaea, Galba, Pseudosuccinea, Austropeplea* and *Radix* snail species.

| Author (Reference) | Objective | Region | Country | Climate | Snail Species Studied | Methods | Outcome |
|---|---|---|---|---|---|---|---|
| Dinnik and Dinnik [32] | To ascertain the impact of seasonal temperature variations on the growth of *Fasciola gigantica* in *Radix natalensis*. | Africa | Kenya | Tropical | *Radix natalensis* | Laboratory experiment | i. Cercariae shed when temperatures rise above 16˚C. ii. The majority of cercariae shed at temperatures between 20–26˚C. |
| Hodasi [33] | To determine the effect of low temperature on *Galba truncatula* | Africa | Ghana | Tropical | *Galba truncatula* | Laboratory experiment | Both juvenile and adult snails survived at 5˚C. ii. Below 5˚C, reproduction was inhibited. iii. Young snails became active at 15–22.5˚C, and shell length rose to 9.22mm. |
| Hoefnagel and Verberk [34] | To investigate the long-term and short-term effects of temperature and oxygen on metabolism, food intake, growth, and heat tolerance in a freshwater snail. | Europe | Netherlands | Temperate | *Lymnaea stagnalis* | Laboratory experiment | i. Snails reared under warm conditions grew faster and consequently reached a larger size than snails reared under cold conditions. ii. Food consumption increases with the temperature rise. |
| Prinsloo and van Eeden [35] | To determine the effect of temperature and its bearing on the distribution and chemical control of freshwater snails. | Africa | South Africa | Tropical | *Radix natelensis* and *Bulinus tropicus* | Laboratory experiment | i. At 25˚C, *R. natalensis* hatched within 14–26 days of egg laying. ii. Best survival is observed at temperatures below 25˚C. iii. At 30˚C, all snails died during the first two weeks. |
| Malone et al. [36] | To establish a geographic information system fasciolosis risk assessment model for East Africa for both *F. hepatica* and *F. gigantica*, which are known to cause significant financial losses in livestock. | Africa | East Africa (Ethiopia, Eritrea, Sudan, Somalia, Kenya, and Uganda) and Djibouti) | Tropical | *Galba truncatula* and *Radix natalensis* | Modelling | i. Altitudes less than 1200 m and an upper limit temperature of 23˚C as conditions for the occurrence of *F. hepatica* |
| Al-Habbib and Al-Zako [37] | To assess the effect of different temperatures on the development of intra-molluscan stages of *Fasciola gigantica* | Asia | Iraq | Dry Desert/Tropical | *Radix auricularia* | Laboratory Experiment | i. Changes in temperature from 15–30˚C reduced the sporocyst duration from 21 to 4 days, the redia duration from 37 to 11 days, the daughter redia duration from 53 to 22 days, and the cercaria shedding from 73 to 25 days. ii. At 12˚C, the parasite progressed to the redia stage in 51 days. iii. The highest cercaria output/snail was recorded at 15˚C and the lowest at 30˚C. |

*(Continued)*

**Table 1.** (Continued)

| Author (Reference) | Objective | Region | Country | Climate | Snail Species Studied | Methods | Outcome |
|---|---|---|---|---|---|---|---|
| Shalaby et al. [38] | To determine how temperature, the severity of the infection, and the age of the *F. gigantica* snail host affect the dynamics of metacercarial productivity. | Africa | Egypt | Tropical or dry desert | *Lymnaea cailliaudi* | Laboratory experiment | i. A temperature of 24°C or higher was ideal for high metacercarial production.<br>ii. The age of snails, not the number of miracidia to which they were exposed to, significantly affected the prepatent duration of *F. gigantica* inside its snail host, which was inversely related to temperature. |
| Singh et al. [39] | To determine whether there was a relationship between abiotic factors and the prevalence of *Fasciola gigantica* infection and how those factors affected the reproduction of vector snail *Radix acuminata* | Asia | India | Tropical monsoon | *Radix acuminata* | Laboratory experiment | i. Uninfected snails have shown higher fecundity at high temperatures (16–34°C) than infected snails do, albeit for a shorter duration of exposure. |
| Jigyasu and Singh [40] | To ascertain the effect of environmental factors on the fertility, hatchability, and survival of juvenile *Radix acuminata* snails. | Asia | India | Tropical monsoon | *Radix acuminata* | Laboratory experimental | i. A temperature range of 16–37°C) of water was associated with snail fertility, hatchability, and survival. |
| Misra and Raut [41] | To determine the effect of temperature on fecundity, growth, and survival of *Radix acuminata* | Asia | India | Tropical monsoon | *Radix acuminata* | Laboratory experiment | i. The rate of growth increased as the temperature increased from 15–30°C but decreased when the temperature exceeded 30°C due to desiccation.<br>ii. At lower temperatures (15–20°C), the reproduction rate was high.<br>iii. Survival rates drop at 25–30° C. |
| Aziz and Raut [42] | To study the effect of temperature on the snail *Radix luteola* under varying environmental conditions such as food, water pH, and salinity (NaCl). | Asia | India | Tropical monsoon | *Radix luteola* | Laboratory experiment | i. Body weight increased with the increase in temperature from 10–35°C.<br>ii. Growth rates in shell length and shell width are noted in snails maintained at 10–30°C.<br>iii. Failed to complete the life cycle at 10–15°C.<br>iv. Maximum reproduction rate at 30°C<br>v. Room temperature has the biggest effect on the rate of egg production, the time required for the development of an embryo, and the hatchability of eggs. |

(*Continued*)

**Table 1.** (Continued)

| Author (Reference) | Objective | Region | Country | Climate | Snail Species Studied | Methods | Outcome |
|---|---|---|---|---|---|---|---|
| Boray [43] | To determine the effect of temperature on fecundity, growth, and survival of *Austropeplea (Lymnaea) tomentosa* | Australia | Australia | Desert or semi-arid coastal corners have a temperate climate, such as oceanic and humid subtropical climate on the east coast | *Austropeplea (Lymnaea) tomentosa.* | Book (experimental work) | i. It was observed that the critical temperature for the life cycle of *F. hepatica* is 9.5˚C. ii. Below 10˚C, there is no activity, or very little, in the parasite larval development and the multiplication of snails. iii. The snail produces eggs well at 25˚C. |
| Harris [44] | To ascertain how temperature affects *Pseudosuccinea columella* and *Austropeplea (Lymnaea) tomentosa* reproduction, growth, and survival. | Australia | New Zealand | Subtropical to subantarctic | *Pseudosuccinea collumela* and *Austropeplea (Lymnaea) tomentosa* | Laboratory experiment | i. *Pseudosuccinea columella's* reproduction rate was high at 24˚C. ii. Eggs from both species need a temperature range of between 5–10˚C to mature and hatch. iii. While developing at 5˚C, some eggs did not survive hatching. iv. *A (L). tomentosa's* eggs were killed by temperatures above 30˚C, while *P. columella* eggs developed normally at 34.5˚C but died at 36˚C. |
| Gold and Goldberg [45] | To determine the effect of temperature on the susceptibility of four snail species to infection with *Fasciola hepatica* (Trematoda), | Asia | Israel | Mediterranean (sub-tropical) climate and temperate (hills) | *G. truncatula, G cubensis, A (L). tomentosa* and *P. columella* | Laboratory experiment | i. It was observed that the infection rate rose with increasing temperatures up to 24˚C and diminished with a further elevation in temperature. ii. It was highest in *G. truncatula* between 16–24˚C, in *P. columella* at 20˚C, and in *A (L). tomentosa* and *G. cubensis* at 24˚C. |
| Robberts [46] | To conduct field research on *Galba truncatula and* its parasite *F. hepatica*, as well as to keep a strain of *G. truncatula* in the lab for experimental purposes. | Europe | Britain (UK) | Temperate | *Galba truncatula* | Field and Laboratory Experiments | ii. They observed 10˚C as a critical temperature for the development of *Fasciola hepatica*. ii. The optimal temperature for growth and development was 25˚C. |
| Pantelouris [47] | To determine the effect of environmental influences on the life cycle of the liver-fluke, *Fasciola hepatica* | Europe | Ireland | Temperate | *Galba truncatula* | Laboratory experiment | i. They observed 10˚C as a critical temperature for the development of *Fasciola hepatica*. ii. The optimal temperature for growth and development was 25˚C. iii. They also noted that there was no larval development below 5˚C. |

(*Continued*)

**Table 1.** (Continued)

| Author (Reference) | Objective | Region | Country | Climate | Snail Species Studied | Methods | Outcome |
|---|---|---|---|---|---|---|---|
| Nice[48] | To determine the effect of temperature on the growth of intramolluscan stages of *F. hepatica* | Europe | Britain (UK) | Temperate | *Galba truncatula* | Laboratory experiment | i. The maximum growth rate was obtained at 20–25˚C. ii. Below 10˚ C, no growth was observed. |
| Lam and Calow [49] | To ascertain how temperature affects Radix (*Lymnaea*) peregra development and survival | Europe | Britain (UK) | Temperate | *Radix (Lymnaea) peregra* | Laboratory experiment | i. Most snails showed higher growth rates and survival rates at 20˚C. |
| Leicht et al. [50] | To examine the adaptation potential of a wider range of life-history traits under climate change conditions. | Europe | Switzerland | Temperate | *Lymnaea stagnalis* | Laboratory experiment | i. They discovered that at heat wave temperatures, snails grew larger and reproduced more, while their immune defences were compromised. ii. Snails exposed to 25˚C grew larger at the end of the study than snails kept at 15˚C. |
| Wilson and Taylor [51] | To assess the influence of variations on *Galba truncatula* parasitization. | Europe | Britain (UK) | Temperate | *Galba truncatula* | Laboratory Experiment | i. The parasite does not develop below 10˚C. ii. Below 5˚C, parasite eggs develop more slowly than snail eggs at temperatures above 10˚C. |
| Venturini [52] | To determine the effect of summer temperature on the *Fasciola hepatica* life cycle in the snail host | South America | Argentina | Temperate | *Lymnaea viatrix* | Laboratory experiment | i. Optimal growth of snails observed at 25.5˚C (21–34˚C) ii. cercaria shedding occurred at 10˚C. |
| AL-Habbib and Granger [53] | To determine the effect of constant and changing temperature on the rate of development of the eggs and the larval stages of *Fasciola hepatica* | Europe | Britain (UK) | Temperate | *Galba truncatula* | laboratory experiment | i. At 10˚C, several generations of daughter rediae were detected. ii. No cercariae shedding at temperatures below 10˚C or over 25˚C. iii. Shedding was observed at 10–25˚C. |
| Kendall and McCullough [54] | To determine the factors that influence the actual emergence of cercariae that are already mature and free in the body cavity of the snail. | Europe | Britain (UK) | Temperate | *Galba truncatula* | Laboratory experiment | i. Observations have shown that below 10˚C, the development of all intra-molluscan stages of *F. hepatica* is inhibited. ii. The critical minimum and maximum temperatures for cercariae shedding were observed to be 10–26˚C, respectively. |
| Souza et al. [55] | To investigate invertebrate host-parasite interaction and assist in the development of an anti-helminthic vaccine. | South America | Brazil | Temperate | *Pseudosuccinea columella* | Laboratory experiment | i. Maximum cercariae shedding was observed at 23–25˚C. |

*(Continued)*

**Table 1.** (Continued)

| Author (Reference) | Objective | Region | Country | Climate | Snail Species Studied | Methods | Outcome |
|---|---|---|---|---|---|---|---|
| Lee et al. [56] | To produce *Fasciola hepatica* metacercaria from its intermediate host, *Lymnaea viridis* | Asia | South Korea | Temperate | *Lymnaea viridis* | Laboratory experiment | i. Between 15–24˚C, snail growth increased. At temperatures above 20˚C, substantial metacercaria production was observed. |
| Vaughn [57] | To determine the effects of Temperature on the Hatching and Growth of *Lymnaea stagnalis* | North America | USA | Temperate | *Lymnaea stagnalis* | Laboratory experiment | i. For 12 days, eggs held at low temperatures (2.4–3.0˚C) produced dead embryos. ii. *Lymnaea stagnalis* eggs require temperatures ranging from 9.9–28.0˚C to mature and hatch. iii. Most eggs failed to hatch at 28.0˚C. |
| Waal [58] | To determine the effects of temperature on the development of *Fasciola hepatica* larval forms in *Pseudosuccinea columella* | North America | USA | Temperate | *Pseudosuccinea columella* | Laboratory experiment | i. No larval development was observed at temperatures below 15˚C. ii. Cercariae shedding occurred between 20–25˚C. When temperatures hit 30˚C and higher, there is a significant increase in larval stage mortality. |
| Abrous et al. [59] | To assess how snail infection by *Fasciola hepatica* and *P. daubneyi* is affected by temperature changes | Europe | France | Temperate | *Galba truncatula* | Laboratory experiment | i. Shedding of cercariae was observed to be slow at low temperatures (6–8˚C) for a short period longer (67±69 days. ii. There was higher Cercaria shedding in snails infected by *Fasciola hepatica* than those infected by *P. daubneyi* at 20˚C. |
| Kendall [60] | To determine the life-history traits of *Galba truncatula* | Europe | Britain (UK) | Temperate | *Galba truncatula* | Laboratory experiment | i. Fecundity has been shown to occur in experimental settings at temperatures as low as 10–11˚C. |
| Claxton et al. [61] | To assess the effects of variable temperatures on *Lymnaea viatrix* snail development and *Fasciola hepatica* parasitism. | South America | Peru (in Cajarmaca) | warm and temperate. | *Lymnaea viatarix* | Laboratory Experiment | i. It was shown that the growth rate increased with temperatures between 10–25˚C. ii. The growth rate was at its highest at 25˚C and began to decrease at higher temperatures. |
| Relf et al. [62] | To highlight the influence of milder temperatures and wetter conditions on both snail and fluke development. | Europe | Ireland | Temperate | *Galba truncatula* | Modelling | i. Temperature is particularly influential in the development of *G. truncatula* and *F. hepatica* because temperatures ≥10˚C are required for growth, reproduction, and survival. |

(*Continued*)

**Table 1.** (Continued)

| Author (Reference) | Objective | Region | Country | Climate | Snail Species Studied | Methods | Outcome |
|---|---|---|---|---|---|---|---|
| Rapsch et al. [63] | To construct an interactive map that depicts the relative risk of transmission by simulating the environmental circumstances that support the survival and reproduction of *F. hepatica* intramolluscan stages and their intermediate hosts. | Europe | Switzerland | Temperate | *Galba truncatula* | modelling | i. The fascioliasis risk is low at temperatures as low as 10°C since egg development takes up to 6 months. ii. Because development takes only 2–3 weeks at temperatures ranging from 23–26°C, the risk in this temperature range is high. iii. The highest risk for egg development was defined at 30°C, where development takes just 8–10 days. iv. The risk decreased rapidly because eggs died at temperatures exceeding 30°C. |
| Hope-Cawdery [64] | To predict the effect temperature on survival of extra-mammalian stages of liver fluke | Europe | Britain (UK) | Temperate | *Galba truncatula* | modelling | i. predicted that the development of Fasciola stages occurs at daily minimal and maximum temperatures of 10°C and 30°C, respectively. ii. The model shows 16°C as the optimal temperature for the development of intra-molluscan stages of *Fasciola hepatica*. |
| Caminade et al. [65] | To simulate recent and future fasciolosis climatic suitability in Europe. | Europe | Britain (UK) | Temperate | *Galba truncatula* | Modelling | i. The model indicated that recent trends are expected to continue in the future with a pattern of climate change for northern Europe, extending the season appropriate for parasite development in the environment. ii. While the disease burden in the south is expected to be lower, |
| **Smith** [89] | To create a mathematical model of Fasciola hepatica's life cycle that, for the first time, incorporates the parasite's seasonality and can be used to forecast parasite abundance under a variety of climate change and farm management practices. | Europe | Britain (UK) | Temperate | *Galba truncatula* | Modelling | i. I t was observed that when temperatures begin to rise, and the eggs start to develop. ii. Warmer temperatures help the snail eggs to hatch into young snails that quickly grow into adult snail. |

**Table 2. Summary of Table 1** shows the IHS of *Fasciola* spp., their preferred climate, the overall thermal tolerance range for each life history trait, and the temperature range for the development of *Fasciola* spp. intramolluscan larval stages.

| Climate | IHS species | Temperature range for IHS growth | Temperature range for IHS reproduction | Temperature range for IHS survival | Temperature range for the development of *Fasciola* spp. intramolluscan stages |
|---|---|---|---|---|---|
| Temperate | *Galba truncatula, Lymnaea viatarix, Pseudosuccinea columella, Lymnaea stagnalis, Lymnaea viridis* and *Radix* (*Lymnaea*) *peregra* | 5–26˚C. | 9.8–28˚C. | 5–26˚C. | 10–30˚C. |
| Sub-tropical | *Pseudosuccinea columella, Lymnaea stagnalis, Galba truncatula, Galba cubensis,* and *Austropeplea* (*Lymnaea*) *tomentosa* | 10–26˚C. | 5–34,5˚C. | 18–30˚C. | 15–30˚C. |
| Tropical | *Radix acuminata, Radix auricularia, Radix natalensis, Radix luteola, Pseuduosuccinea columella,* and *Galba truncatula* (in high-altitude or high-land areas of Africa, Latin America, and Asia) | 10–36˚C. | 15–36˚C. | 16–30˚C. | 12–30˚C. |
| **Desert** | *Lymnaea cailliaudi* and *Radix auricularia* | 10–36˚C. | 18–24˚C. | 15–30˚C. | 12–30˚C. |

## Temperature effect on the survival of IHS

Several studies (Table 1) showed that lymnaeid snails do not survive at temperatures below 5˚C [33] and that their survival rate is greatly reduced when temperatures get to 25–35˚C. Temperatures above 30˚C were lethal to developing embryos of *L. tomentosa; P. columella* hatched at up to 34.5˚C, but at 36˚C none survived [66]. In the studies carried out by Aziz and Raut [42], the survival rate of *Radix acuminata* in the tropics decreased between temperatures of 25˚C and 30˚C, while the survival rate for *Lymnaea peregra* in the subtropics was high at 20˚C [49]. In general, IHS survival decreased with increasing temperatures.

## Temperature and parasite development

In an experiment to ascertain how seasonal temperature fluctuations affect the development of *Fasciola gigantica* in *Radix natalensis* in the tropics, it was observed that at 16˚C, rediae developed from the sporocyst [32]. It was noted that cercariae were shed at a minimum temperature above 16˚C, and the majority of the cercariae were shed at a mean maximum temperature of 20˚C [32]. In the temperate region, cercariae shedding for *Fasciola hepatica* took longer at low temperatures (6–8˚C), and the time for the process was significantly reduced at 20˚C in the temperate region [59]. No *F. hepatica* larval development was observed at temperatures below 15˚C in *P. columella* [58]. The maximum development rate of *F. hepatica* inside *G. truncatula* was observed at temperatures from 23–25˚C, and no larval development was observed at temperatures below 10˚C [37,43,48,51,53,55,57]. Another study found that several generations of daughter rediae were observed at 10˚C, that shedding occurred at 10–25˚C, and that no cercariae shedding occurred at temperatures below 10˚C or above 25˚C [46,47,52–54,62]. It was observed in the subtropics that the infection rate rose with increasing temperatures up to 24˚C and diminished with a further elevation in temperature. Infection rates were highest in *G. truncatula* at 16–24˚C, in *P. columella, and* at 20˚C in *A* (*L*). *tomentosa* and *G. cubensis* [67]. Most studies reviewed concluded that the optimal temperature for *Fasciola* development was between 10˚C and 30˚C.

## Temperature-dependent model formulation

Numerous statistical and mathematical models have been developed to understand the dynamics of fascioliasis transmission. Most of these models synthesize information from experimental studies and extrapolate to give future projections.

## Statistical models

The study by Malone et al. [36] showed that 23°C was an optimal temperature for the distribution of *F. hepatica* in the Ethiopian highlands. Prediction of the distribution of snails has been used to potentially indicate the risks of disease transmission. Studies by Stensgaard et al. [68] and Pedersen et al. [69] have combined aspects of temperature and Geographic Information Systems (GIS) to predict the spatial distribution of the disease. These studies indicate areas that will be suitable for both snail and parasite distribution and further suggest that variable habitats may be created due to a temperature rise.

## Mathematical models

A model by Rapsch et al. [63] suggested that the risk for fascioliasis is high at high temperatures. As the development of *Fasciola hepatica* takes only 2–3 weeks at temperatures of 23°C–26°C, risk in this temperature range is high. It was observed that the risk of the disease caused by low land strain of *F. hepatica* is high at altitudes below 1800m and decreases as altitude increases [63]. The month of August was predicted to have a high fascioliasis prevalence in Switzerland as this is the with favourable temperatures for the development of the parasite. Another model by Hope-Cawdery [64] predicted that the effect of temperature on the survival of extra-mammalian stages of liver fluke occurs at a temperature range of 10–30°C. They observed the effect of temperature on survival to the completion of the intra-molluscan stage of *F. hepatica*, with an optimum at about 16°C. It also shows that, at low temperatures, the combination of long development times and increasing mortality rates results in a rapidly falling potential for transmission. This occurs in nature, as the presence of snails in Iceland without transmission demonstrates. Most models identified rainfall and temperature as the most important factors for the survival and development of the parasite's free-living stages, as well as the IHS [65].

## Discussion

### Temperature effect on fecundity of IHS

Generally, most living organisms, including snails, channel most of their resources towards their life history traits (growth, fecundity, and survival) [28]. The predicted rise in temperatures will have an effect on snail fecundity and egg development; and this will have important implications on the future distribution and population dynamics of the IHS [66]. The results of this review study suggest a positive association between temperature and snail fecundity. The study has shown that there will be a likely increase in snail population within favourable temperature range [33,50] while, rising temperatures beyond the favourable temperature range due to climate change may negatively affect the life-history traits of snails. This effect on life-history traits will potentially lead to a reduction in egg viability and or reduced survival due to changes in the allocation of resources [70]. Temperature also influences organisms' physiological functions by altering the metabolic rates and aerobic scope of ectotherms through thermodynamic effects [71]. The concept of energy allocation maybe the case. At low temperatures, snails may not feed as much thus, energy reallocation is towards survival than reproduction [33]. According to Khallaayoune et al. [72], cooler temperatures may also affect the synchronization of reproductive events such as the timing of mating and egg-laying. Furthermore, these factors might limit the reproductive success and population growth of lymnaeid snails in temperate regions. In contrast, the results of the reviewed papers indicated that temperatures ranging from 10–25°C enhanced egg mass production in *Galba truncatula* [73]. Most studies suggested that within favourable temperature range, the reproductive activities of

lymnaeid snails can be stimulated, leading to higher fecundity [40,42]. However, most studies also indicated that temperatures above 25˚C have negative adverse effects on the fecundity and fertility of snails from temperate regions [44,66]. Heat wave temperature at 25˚C was associated with *Lymnaea stagnalis* producing more eggs, while their immune defence was reduced [50]. The reviewed articles showed that IHS from temperate, Oceania, the Mediterranean, and subtropical regions are temperature sensitive [57,60,66]. On the other hand, IHS from the tropical regions produce more egg masses in the low-temperatures, while the opposite is true in high temperatures for those observed in the tropical regions [40–42]. Higher temperatures might have a positive effect on the reproductive events in lymnaeid snails. For example, *Radix acuminata* fecundity was optimal at warmer temperatures, between 15˚C and 36˚C [40–42]. The variations observed in optimal temperatures for fecundity vary with the snail species, size, age, and experimental protocol used. The production of egg masses at different temperatures might suggest that some IHS species are well adapted to low- or high-temperature ranges [28]. This was noted in *Pseudosuccinea columella*, which has adapted to subtropical and tropical climates. Most of the reviewed articles in this study on lymnaeid snails suggested that most snails' fecundity ceased at temperatures below 5˚C and over 30˚C [33,46]. To guide snail control activities, it is critical to understand the optimal temperature range for fecundity for lymnaeid snails and the predicted effect of climate change on snail population dynamics.

## Temperature effect on the growth of IHS

Snail growth is another important life history trait that can influence snails to invade new places and adapt to new environments. Growth affects fecundity, ability of IHS to resist trematode infections, and resource availability to the development of intramolluscan larval stages of the trematodes [28]. Snails maintained at 25˚C were observed to be larger compared to snails maintained at 15˚C [50]. Also, development rates are more temperature-sensitive for species from cooler habitats and terrestrial species than those from warmer habitats and aquatic habitats [70]. This is because warmer temperatures increase feeding habits and metabolic processes, which generate energy for the growth of IHS. Growth has been observed to be influenced by temperatures between the lower threshold and upper threshold temperatures (Table 2). For example, the temperature range of 10–26˚C for *Galba truncatula* [60]. Below and above these temperatures, snail growth may be inhibited while snail mortality increases. For instance, Heppleston [74] suggested that growth was most rapid in the warm summer months, which incidentally is the period when snail mortality was greatest. For example, the optimal growth rate temperature for *Lymnaea viatrix* snails was 25˚C, and above this temperature, the growth rate declined [61], while Venturini [52] observed the optimal growth rate temperature for the same species at 25.5˚C. Warmer temperatures and low altitudes have been observed to favour the proliferation of *Radix natalensis* over *Galba truncatula* in the tropical regions of Africa and Asia [8,36]. This observation may suggest that climate change may result in the shift in the transmission dynamics of *Fasciola hepatica* infections by *Pseudosuccinea columella* which is associated with colonisation of fresh water habitats [23]. At extreme temperatures, the growth of snails declines, and death rates increase. Most of these studies showed that high temperatures within a suitable thermal threshold range promote snail growth. For example, the range of temperatures in tropical locations is 15–35˚C; in temperate zones, it is 10–26˚C; and in desert or arid regions, it is 18–24˚C. A temperature-driven high growth rate might result in increased fecundity, higher survival rates, and the potential of the vector snails to spread fascioliasis to new areas.

## Temperature effect on the survival of the IHS

The disease transmission cycle cannot be completed without the existence and survival of a compatible host [28]. The survival time of IHS is influenced by both biotic and abiotic factors. These factors include the species of the snail, habitats types, soil moisture and relative humidity, and tolerance for low- or high-altitude habitats [75]. For example, *G. truncatula* snails in the Andean highlands of South America adapt to the extreme environmental characteristics of very high-altitude areas [16,76,77]. In Africa, *Galba truncatula* has adapted to cooler, highlands or high-altitude areas in Lesotho, South Africa [78], North African nations (Morrocco, Egypt, Tunisia), and East African nations (Ethiopia, Kenya, Uganda, Tanzania) [36], and this is the case in the high lands or altitude in Bolivia [2,77,79,80], which has enhanced the spread of *Fasciola hepatica* infections. These adaptation techniques include having a longer cercarial shedding period, increased cercarial production per snail, and prolonged survival of infected snails [76]. The absence of *Fasciola gigantica* in South America ensured fascioliasis transmission by *Galba/Fossaria* species [14].

Temperature may influence IHS survival; it has been observed that snails survive longer in low-temperature environments than in high-temperature habitats [73]. It has been shown that IHS exhibit phenotypic modification after infection, which affects their life span [28]. Snail survival is one of the primary requirements for trematode development and the maturation of their intramolluscan larval stages. Several IHS are observed to have different optimal temperatures to survive. For example, *Radix natalensis*, which is the main IHS for *Fasciola gigantica* in tropical and subtropical habitats, can tolerate warm temperatures in the range of 18–25°C compared to *Galba truncatula*, which is the IHS for *Fasciola hepatica* in temperate regions and prefers a cooler environment (10–25°C) [81]. On the other hand, high temperatures have also been associated with altered membrane permeability, increased accumulation of harmful metabolites, and weakened immune systems in snails [28]. This may decrease the snails' survival chances by making them more susceptible to infection. Despite this, *G. truncatula* has developed an overwintering ability to withstand extremely cold environments and aestivation in unstable water bodies to resist drought periods [16,46,82]. Therefore, the survival rates of most lymnaeid snail spp. are inversely correlated to an increase in temperature.

Rising temperatures and habitat microecology factors such as soil surface temperature, air temperature, soil humidity, and soil and water pH in these regions might drive lymnaeid snails to shift their geographic distribution [4,25]. Cooler regions that were once suitable for these temperate regions may become less favourable, and extremely cold regions in the temperate regions that were previously unsuitable may become more habitable [24,77]. This can lead to changes in the distribution and abundance of lymnaeid snails, potentially affecting the overall disease transmission dynamics [65]. However, most of the studies were done on infected snails. Understanding the thermal tolerance range of both infected and non-infected snails may be important in predicting the overall effect of climate change on fascioliasis.

## Temperature effect on the development of intramolluscan stages of *Fasciola spp*

The development of the parasite larval stages within snail hosts is temperature-dependent [83–85]. This was supported by the results of the reviewed articles in Table 1, which suggested that high temperatures within a certain favourable thermal regime by IHS promote the rapid development of *Fasciola* spp. intramolluscan larval stages and the emission of cercariae. Therefore, it was suggested that high temperatures shorten the prepatent stage of *Fasciola* spp. [12,54,60,73,85]. Temperatures below and above the optimal temperature may result in a decrease in parasite development. Furthermore, Table 2 presents the optimal temperatures for

the development of intramolluscan trematode stages for each of the IHS studied. These variations may result from differences in the protocols or experimental conditions, the ability of the IHS to tolerate both warmer and colder temperatures, and many other compounding factors like the type of snail food, the mammalian source of the parasite eggs, the altitude of the isolates' origin, and variations in daily and seasonal temperatures [85–87]. Despite the variations in optimal temperatures for the development of *Fasciola spp* inside their IHS, it was observed that the development rates of helminth eggs and early larval stages increase with rising temperatures below the maximal development threshold, of which temperatures above this might stop parasite development [61]. However, the risk of disease transmission decreases at temperatures above 30˚C, as IHS dies at this thermal regime due to desiccation [61,63]. It was further noted that the development of *Fasciola spp* within their intermediate hosts might be affected by the species involved and by the rate at which they adapt to the new environment created by climate change. There was concordance between the presence of *G. truncatula* or *F. hepatica* and *R. natalensis* or *F. gigantica*, suggesting that the determining factor in the distribution of *F. hepatica* and *F. gigantica* is not temperature *per se* but rather the presence of an intermediate host in which each species can reproduce efficiently [88]. Additionally, the development of parasites within a host is significantly impacted by the strong immunological response of IHS against Fasciola infections. For example, *Pseudosuccinea columella* in Cuba has been found to have some immune response or resistance against *F. hepatica* infections [22].

## Disease models

Several statistical models have been developed in the past to predict how climatic factors, specifically temperature and rainfall, affect parasite abundance [24]. As the temperature at which infection is carried out increases, there is a proportional increase in parasitization of IHS [51]. However, these models were developed under historic climate conditions or for a specific geographic location, and thus are not well suited to predicting how infection epidemiology might shift in situations of global climatic change [89]. Hence, an answer to it is the development of a mechanistic model. To develop it, data from the fieldwork must be combined with the data from experimental studies on the effect of temperature on host-parasite interactions.

## Statistical models

Statistical models have been utilized to better understand the risk factors for fascioliasis and to discover correlations between various disease transmission factors. These models identify optimal areas for snail and parasite distribution. The results also imply that when temperatures rise, variable ecosystems may emerge. Furthermore, regression techniques have been regarded as useful to species distribution modeling since they are concerned with demonstrating correlations between the dependent variable, which can be binary, counts, or ordinal, and the independent or explanatory variables [90,91]. Regression approaches have been utilized in the research of fascioliasis to explain the geographical variation of infections throughout several areas of investigation [92]. The use of geographic information systems (GIS) and remote sensing (RS) technology in epidemiological research and the control of zoonotic diseases has increased over the last few decades [93]. Fascioliasis is a suitable candidate for the application of RS and GIS in epidemiology because it is strongly influenced by the environment, particularly the habitat of the intermediate host. Temperature is an essential element that might limit the development of fascioliasis and the creation of new snail habitats, according to models developed by Pedersen et al. [69]. Further iterative GIS analysis by Malone et al. [36] suggested that soil acidity regime in Southwestern Sudan and Uganda (<5.5 pH) may be unsuitable for lymnaeid snail hosts of *F. hepatica* and or those tropical thermal regimes with an average

annual mean temperature above 23˚C, as unsuitable for *Galba truncatula*. The average annual mean temperature of 23˚C or above was found to correspond to areas below the reported 1200 m elevation limit of *F. hepatica* in Ethiopia [36].

## Mathematical models

Rapsch et al. [63] and Fox et al. [24] predicted that changes in the global temperature may have an impact on both the snails' geographic distribution and the abundance of *Fasciola spp*. However, the research by De Kock et al. [78] shows that temperature, altitude, type of water-body, and, to a lesser extent type of substratum, are critical factors that may greatly affect the geographical distribution of *G. truncatula*. Although models based on temperature and rainfall have been developed to estimate fluke abundance, these models usually use data from the season during which the transmission is known to occur [89]. These models are not reliable in predicting changes in the seasonality of infection. Most previous models identified rainfall and temperature as the key drivers in the transmission of fasciolosis [94,95]. However, current models incorporate GIS and remote sensing (RS) in mapping the suitable habitats for IHS and the distribution and abundance of IHS species in a particular area. The highest risk for the development of eggs was defined at 30˚C where development only takes 8–10 days. However, the risk of the disease transmission decreased at a temperature above 30˚C, as eggs die at this thermal regime. Rapsch et al. [63] observed that at low temperatures like 10˚C the fascioliasis risk is low because the development of the eggs takes up to 6 months. In another case, a model predicted that the occurrence of *G. truncatula* decreases with the increase in temperature, whilst the distribution of *R. natalensis* increases with the rising temperature, but within its thermal tolerable range [23]. The main key drivers of fascioliases in the global database were rainfall and potential evapotranspiration across the entire globe [24,96]. As a result, the resulting equation for generating the index was modified to allow for the assessment of fascioliasis risk from either *F. hepatica* or *F. gigantica*, as both species thrived in East Africa [36]. In East Africa, risk index maps were developed for both species. The computed risk score showed geographic variation in the transmission patterns of both fascioliasis species. Places with higher soil moisture content were riskier than those with lower soil moisture content [36].

## Potential future research areas

The impact of climate change on fascioliasis is complex due to the multitude of factors that may be at play [97]. While temperature can influence the spread of fascioliasis, the availability of food, parasite, compatible host, the ability of the pathogen to utilize a variety of host snails [11,98], and water sources for the snails and the presence of other climatic and human-related factors may also play a role [99]. For instance, higher humidity levels and changes in water availability could modify the transmission dynamics and spread of the parasite [6]. Furthermore, changes in host species composition may cause the disease's natural distribution to expand or constrict. Overall, the potential impact of climate change on fascioliasis is uncertain, and more research is needed to fully predict the effects of climate change on fascioliasis transmission. Furthermore, more research into the impact of the various factors that may be involved should be conducted to gain a better understanding of the complex transmission dynamics of fascioliasis.

There is a need to understand the effect of temperature on all the development stages of the snail-parasite system as well as the survival rates of IHS over time [83,100]. The tolerance of successive snail generations at various temperature regimes may also need to be assessed. This is because the parameters employed in disease models based on the performance of a single generation may change as snails adjust to greater temperatures over time. To achieve this, the

use of experimental data coupled with field studies may prove valuable in observing the changes in the environment. Temperature-driven experiments based on infected and non-infected snails may assist in understanding the life history trials and disease transmission dynamics of fascioliasis and other snail-borne diseases that may be affected most by climate-driven rise in temperature. The use of geographic information systems (GIS) in models will also assist in projecting the areas that can be future disease hot spots owing to the rise in temperature. Furthermore, this review revealed that many studies on fascioliasis have focused on *F. hepatica*, which thrives in temperate climates and subtropical developed countries, and less on *F. gigantica*, which is most prevalent in tropical regions of the developing world. The review also exposed that few experimental studies explore the effect of temperature on life history traits of fascioliasis intermediate host snails in Africa. No mechanistic model has been developed to forecast how future temperature increases will affect the dynamics of fascioliasis transmission in Africa.

## Conclusion

The review has shown that temperature is a key determinant factor in fascioliasis transmission. It also shows that it may continue to play an essential role in developing fascioliasis control programs. Temperature increases due to climate change might lead to a complex interaction between IHS and *Fasciola*. Although a temperature rise may most likely lead to the disappearance of fascioliasis in certain areas, a possible increase in the snail population due to a rise in fecundity and a reduction in the parasite development rate may increase *Fasciola* infective stages in other areas. Studies have shown that *Galba truncatula* is a major IHS for *F. hepatica* in temperate, subtropical, and tropical highlands regions across the globe. The ability of *P. columella* to adapt to cooler, warmer, and higher temperatures, and to harbour both *Fasciola* species enables it to be a major transmitter of fascioliasis. Mathematical and statistical disease modelling may however be a useful tool in understating the epidemiology of fascioliasis. For better predictions, analysis, and application, these models might need to include temperature-dependent stages as well as the biology of IHS and *Fasciola spp*. The precision of models will also be increased by using data from well-designed laboratory and field studies that systematically assess the impact of temperature at various stages of snail and parasite development.

## Acknowledgments

We acknowledge the College of Health Sciences at University of KwaZulu-Natal for providing the first author (AD) with fee remission for his PhD studies.

## Author Contributions

**Conceptualization:** Agrippa Dube, Chester Kalinda, Moses John Chimbari.

**Data curation:** Agrippa Dube.

**Funding acquisition:** Moses John Chimbari.

**Investigation:** Agrippa Dube.

**Methodology:** Agrippa Dube.

**Project administration:** Moses John Chimbari.

**Resources:** Moses John Chimbari.

**Supervision:** Chester Kalinda, Moses John Chimbari.

**Validation:** Moses John Chimbari.

**Writing – original draft:** Agrippa Dube.

**Writing – review & editing:** Agrippa Dube, Chester Kalinda, Tawanda Manyangadze, Tafadzwa Mindu, Moses John Chimbari.

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
