## [Decision Letter · Decision Letter 0]

10 Oct 2023

Dear Mr Dube,

Thank you very much for submitting your manuscript "Effects of temperature on the life history traits of Intermediate Host Snails of Fascioliasis : A Systematic Review" for consideration at PLOS Neglected Tropical Diseases. As with all papers reviewed by the journal, your manuscript was reviewed by members of the editorial board and by several independent reviewers. In light of the reviews (below this email), we would like to invite the resubmission of a significantly-revised version that takes into account the reviewers' comments. 

We cannot make any decision about publication until we have seen the revised manuscript and your response to the reviewers' comments. Your revised manuscript is also likely to be sent to reviewers for further evaluation.

Revision of the structure and language throughout is critical to improve this manuscript, as noted by both reviewers.

Sincerely,

María Victoria Periago

Academic Editor

Victoria Brookes

Section Editor

The submitted manuscript discusses the impact of climate change on the biological processes which affect the transmission dynamics of snail-borne diseases, especifically Fascioliasis, through a review on experimental and model studies. The reviewers agree on the relevance and importance of this study but have made some comments and suggestions to improve the manuscript and clarify some areas, therefore we suggest taking them into consideration.

Reviewer's Responses to Questions

**Key Review Criteria Required for Acceptance?**

**Methods**

-Are the objectives of the study clearly articulated with a clear testable hypothesis stated?

-Is the study design appropriate to address the stated objectives?

-Is the population clearly described and appropriate for the hypothesis being tested?

-Is the sample size sufficient to ensure adequate power to address the hypothesis being tested?

-Were correct statistical analysis used to support conclusions?

-Are there concerns about ethical or regulatory requirements being met?

Reviewer #1: (No Response)

Reviewer #2: The objectives are well defined and easy to understand. The study methodology is well described, in a structured way and meets the proposed objectives. All inclusion and exclusion criteria are well defined, and well complemented with the figure. Given that this is an analysis of published articles, the sample size depends on the information available in the literature.

**Results**

-Does the analysis presented match the analysis plan?

-Are the results clearly and completely presented?

-Are the figures (Tables, Images) of sufficient quality for clarity?

Reviewer #1: (No Response)

Reviewer #2: The results were well presented and well structured, although the English is very confusing, which makes interpretation difficult.

The table is quite illustrative of the results obtained from the various articles analyzed.

**Conclusions**

-Are the conclusions supported by the data presented?

-Are the limitations of analysis clearly described?

-Do the authors discuss how these data can be helpful to advance our understanding of the topic under study?

-Is public health relevance addressed?

Reviewer #1: (No Response)

Reviewer #2: The results are well discussed, and well organized, presenting a great reflection on the importance of this type of studies. The conclusions are written in an organized manner and reflect the results obtained. In addition to the results obtained, limitations of the study are also presented, which adds value to the work.

**Editorial and Data Presentation Modifications?**

Reviewer #1: (No Response)

Reviewer #2: To improve the quality of the manuscript, there are some issues that should be revised, namely:

Abstract

Line 26 – you say Intermediate Host Snails (IHS) – Should maintain the same criteria during the manuscript (Ex: Lines 36,37 and 38)

Background

Line 66 – liver fluke is the parasite, not the disease;

Line 71 – sporocysts, rediae (not radiae) and cercariae are intramolluscan stages of trematodes, not intramolluscan tematodes

Line 75 – Galba truncatula (also called Lymnaea truncatula)

Line 75 – temperate regions

Line 76 – Galba natalensis (also called Lymnaea natalensis) – not Lymnaea truncatula

Line 75 – Pseudosuccinea columella (also called Lymnaea columella)

Line 88 – presence of the vector snails and parasites

Material and Methods

Line 117 – You should explain why the search by Lymnaea, Galba, Pseudosuccinea and Radix. Maybe you should also search for Fossaria

Line 118 – Galba, Radix (italics)

Line 126 – Fasciola hepatica (italics)

Line 131 – If you say Lymnaea sp, must say Galba sp and Pseudosuccinea sp

Table 1. – Revise the italics; inoculated miracidia??? (page 12); Sometimes you say “x - yºC”, other "xºC - yºC" (should use the same criteria);

Line 175 – Temperature is known to "have" impact "on" IHS growth 

Line 215 – Studies Stensgaard et al (???????)

225 – snail intermediate hosts – shoud be SIH 

Lines 242, 247 – sometimes is Lymnaeid, other lymnaeid (it is not the only case)

Line 341 – intramolluscan stages of Fasciola hepatica

Line 374 – parasitization (????)

Line 421 – GIS (not a Geographic Information System (GIS))

References

Line 500 – [5] incomplete

Line 502 – [6] Tidman, R., et al… … (must see the recommended referencing style) You have much more cases like this.

Line 516 – [11] incomplete

Line 537 – [20] incomplete

Line 540 – [22] incomplete

Line 657 – [67] incomplete

Line 702 – [86] “Note: MDPI stays neutral with regard to jurisdictional claims in published …” (????)

**Summary and General Comments**

Reviewer #1: (No Response)

Reviewer #2: Despite being a very relevant topic, it has some limitations. It is written in a very confusing way, difficult to understand, specially at the background seccion. It is used different writing criteria throughout the manuscript, with many errors in terms of italics in the genera and species of parasites/molluscs, which requires extensive revision. There is a lot of inconsistency in the way the data is presented, and several errors in the formatting of bibliographic references.

PLOS authors have the option to publish the peer review history of their article (what does this mean?). If published, this will include your full peer review and any attached files.

Reviewer #1: Yes: RONALDO DE CARVALHO AUGUSTO

Reviewer #2: No
---

## [Editor Report · Decision Letter 1]

21 Nov 2023

Dear Mr Dube,

We are pleased to inform you that your manuscript 'Effects of temperature on the life history traits of Intermediate Host Snails of Fascioliasis : A Systematic Review' has been provisionally accepted for publication in PLOS Neglected Tropical Diseases.

Best regards,

María Victoria Periago

Academic Editor

Victoria Brookes

Section Editor

The authors have addressed all of the reviewers comments and have greatly improved the manuscript. Please go over some minor issues when you proof the document for publication. For example, line 73, when you start a sentence with a species name it need to be written out "Fasciola gigantica" not "F. gigantica". Line 157, there is space missing between "Table 1," and "showed".

---

## [Editor Report · Acceptance letter]

29 Nov 2023

Dear Mr Dube,

We are delighted to inform you that your manuscript, " Effects of temperature on the life history traits of Intermediate Host Snails of Fascioliasis : A Systematic Review," has been formally accepted for publication in PLOS Neglected Tropical Diseases.

Best regards,

Shaden Kamhawi

co-Editor-in-Chief

Paul Brindley

co-Editor-in-Chief
